# Preoperative Fibrinogen and Hematological Indexes in the Differential Diagnosis of Idiopathic Granulomatous Mastitis and Breast Cancer

**DOI:** 10.3390/medicina57070698

**Published:** 2021-07-08

**Authors:** Mehmet Velidedeoglu, Berrin Papila Kundaktepe, Hulya Aksan, Hafize Uzun

**Affiliations:** 1Department of General Surgery, Cerrahpasa Faculty of Medicine, Istanbul University-Cerrahpasa, Istanbul 34098, Turkey; m-veli@hotmail.com; 2Department of Biochemistry, Medicine of Faculty, Haliç University, Istanbul 34445, Turkey; hulyaaksan@halic.edu.tr; 3Department of Biochemistry, Cerrahpasa Faculty of Medicine, Istanbul University-Cerrahpasa, Istanbul 34098, Turkey; huzun59@hotmail.com

**Keywords:** idiopathic granulomatous mastitis, breast cancer, recurrence, CRP, albumin, fibrinogen, neutrophil–lymphocyte ratio

## Abstract

*Background and Aim*: Studies on hematological parameters in the differential diagnosis of idiopathic granulomatous mastitis (IGM) and breast cancer (BC) are limited. This study investigated whether preoperative fibrinogen and hematological indexes can be used in the differential diagnosis of patients with IGM and early-onset BC. *Methods*: Fifty patients with BC, 55 patients with IGM, and 50 healthy volunteer women were included in the study. *Results*: There was a statistically significant difference between the IGM and the BC with respect to fibrinogen, fibrinogen/albumin (Fib/Alb) ratio, C-reactive protein (CRP), white blood cells (WBC), neutrophils, neutrophil–lymphocyte ratio (NLR), platelet–lymphocyte ratio (PLR), and monocyte values. When fibrinogen (*p* < 0.001), the Fib/Alb ratio (*p* < 0.001), CRP (*p* < 0.001), WBC (*p* < 0.001), neutrophil (*p* < 0.001), NLR (*p* < 0.001), monocyte (*p* = 0.008), and 2-hour sedimentation rate (*p* < 0.001) were compared between the groups, the highest levels were found in the IGM group. There was a negative relationship between CRP and albumin, and a positive relationship was observed between CRP and WBC, NLR, PLR, and 2-h sedimentation rate. CRP had the highest sensitivity (95%), whereas the Fib/Alb ratio (86%) had the highest specificity. Patients with recurrent IGM had increased fibrinogen, Fib/Alb, CRP, neutrophils, NLR, and 2-h erythrocyte sedimentation rate (ESR) and decreased lymphocyte levels compared to non-recurrent patients. *Conclusions*: Preoperative CRP, albumin, fibrinogen, Fib/Alb, WBC, neutrophil, NLR, monocyte, and 2-h ESR have considerable potential to be early and sensitive biomarkers of IGM caused by inflammation compared to BC. These parameters also have a significant effect on the recurrence of the disease, suggesting their potential as a practical guide for the differential diagnosis of BC from IGM.

## 1. Introduction

Idiopathic granulomatous mastitis (IGM) is a rare, chronic inflammatory breast lesion of unknown etiology that clinically and radiologically mimics breast cancer [1]. IGM, which was first described by Kessler and Wolloch [2] in 1972, although clinically seen in both breasts, often creates a unilateral mass. Especially as the disease progresses, fistulization, withdrawal of the breast skin and nipple, and axillary lymphadenopathy in IGM mimic breast cancer [3]. Although the exact cause of this inflammatory condition is unknown, the disease has been associated with oral contraceptive use, pregnancy history, and breastfeeding, and its treatment remains controversial [4]. The definitive diagnosis is made after the exclusion of other causes of granulomatous mastitis by histopathological examination. Since the differential diagnosis of IGM from breast cancer (BC) cannot be made clinically, it has led to the investigation of non-invasive methods and molecules [5].

BC is the second most commonly diagnosed cancer worldwide, behind lung cancer [6]. The International Agency for Research on Cancer has recently declared that the ratio of new breast and lung cancer cases in 2020 for both sexes was 11.4% and 11.7%, respectively [6]. BC is also the second most common cause of female cancer deaths in the United States [7]. Improvements in both BC screening and adjuvant therapy have provided a decrease in mortality [8,9]. However, therapy increases the ratio of overall survival when breast cancers are treated earlier [10]. Novel diagnostic markers may play a role in the early diagnosis of breast cancer. Inflammation is a critical component of cancer progression [11]. Infection sites, chronic irritation, and inflammation are the sources of many tumors, as inflammatory cells play a role in the neoplastic process and survival [11,12,13]. Cost-effective prognostic models are desired and neutrophil and lymphocyte counts are available in routine blood tests as part of an automated complete blood cell count. There is a relationship between the neutrophil–lymphocyte ratio (NLR) and tumor growth, progression, invasion, and metastasis [11,14].

NLR and fibrinogen levels, which are among the new indicators of systemic inflammation, have been associated with the prognosis of tumors such as gastric cancer [15], colon cancer [16], nasopharyngeal cancer [17], and liver cancer [18]. Recent studies have shown that systemic inflammatory markers are associated with BC prognosis, and most studies have focused on the prognostic role of the NLR [19], platelet–lymphocyte ratio (PLR) [20], and Glasgow prognostic score [21]. NLR has also been reported to be a supportive marker in the diagnosis of inflammatory and autoimmune diseases [22,23].

Studies on hematological parameters in the differential diagnosis of IGM and BC are limited [24]. The current study investigated whether there could be potential new biomarkers to predict the differential diagnosis of IGM and BC by determining preoperative fibrinogen and hematological indexes.

## 2. Materials and Methods

### 2.1. Study Participants

The study was approved by the Ethics Committee of the Cerrahpasa Faculty of Medicine (No: 86674, date: 9 July 2020). Fifty patients with solid tumors who were followed by a diagnosis of BC in the general surgery outpatient clinic of Cerrahpasa Medical Faculty, 55 patients diagnosed with IGM, and 50 healthy volunteer women with similar demographic characteristics were included in the study. Women over the age of 18, whose microbiological culture samples were confirmed to be sterile and found to have cancer and IGM in the histopathological evaluation, were included in the study. We excluded women who received chemotherapy or radiotherapy and were diagnosed with old or new BC or IGM, with another history of malignancy (DM, hypertension, renal or cardiovascular disorders, etc.), and were pregnant or lactating. In addition, according to core needle biopsy taken from the suspicious lesion in the breast or pathological examination of the tissue taken from the abscess wall, patients with tuberculosis capable of granulomatosis reaction, infection such as fungal and parasitic reactions, sarcoidosis, Wegener granulomatosis, and non-infectious agents such as foreign bodies were excluded from the study.

Venous blood samples were collected from control subjects and patients with BC and IGM after 12 h of fasting. The samples were centrifuged for 15 min at 3000× *g*. Plasma and serum were transferred to Eppendorf tubes and was studied immediately.

Demographic data (age, weight, height, body mass index), biochemical parameters, cancer antigens (Carcinoembryonic antigen (CEA), CA 19-9, CA 15-3, CA-125), and pathology reports were obtained from the subjects’ medical records.

Biochemical parameters were measured on an Olympus AU 800 analyzer by enzymatic methods using commercial kits (Roche Diagnostics, GmbH, Mannheim, Germany) at the Central Biochemistry Laboratory of Cerrahpasa Medical Faculty. Tumor markers were measured by immunometric assay using an IMMULITE 2000 (DPC, Los Angeles, CA, USA).

### 2.2. Statistical Analysis

SPSS 21 for Windows was used to conduct the statistical analyses. The Kolmogorov–Smirnov test was used for normal distribution. The Kruskal–Wallis test was used to compare data of three independent groups; adjusted *p* values were used to conduct the post-hoc test after the Kruskal–Wallis test. The Mann–Whitney U Test was used to compare two independent groups. Spearman’s correlation was used to show an association between two continuous variables. Receiver operating characteristic (ROC) curve analysis was performed to determine the sensitivity and specificity values of biochemical and hematological indexes that can be used to define BC and IGM. The statistical significance level was set at *p* < 0.05.

## 3. Results

The demographic characteristics and tumor markers of the study groups are shown in Table 1. Clinicopathological features of the BC group are shown in Table 2.

As shown in Table 3, albumin, hemoglobin, and hematocrit levels were significantly lower in the IGM group compared to the control (*p* < 0.01). The two-hour erythrocyte sedimentation rate (2-h ESR) was significantly higher in the IGM group compared to the control (*p* < 0.01). A comparison of fibrinogen (*p* < 0.001), the fibrinogen/albumin (Fib/Alb) ratio (*p* < 0.001), C-reactive protein (CRP) (*p* < 0.001), white blood cells (WBC) (*p* < 0.001), neutrophil (*p* < 0.001), NLR (*p* < 0.001), monocyte (*p* = 0.008), and 2-h ESR (*p* < 0.001) within the groups revealed the highest levels in the IGM group.

Correlation analysis of the biochemical variables of the IGM group is shown in Table 4. A comparison of the ROC curves with the sensitivity, specificity, area under curve (AUC), cut-off, and asymptotic significance of CRP, fibrinogen, CA125, Fib/Alb ratio and NLR in the IGM are shown in Table 5 and Figure 1.

The recurrence rate of IGM was 20% at 2 years of follow-up. Biochemical laboratory findings and hematological indexes of patients with non-recurrent and recurrent IGM are shown in Table 6. In patients with recurrent IGM, fibrinogen, the Fib/Alb ratio, CRP, neutrophils, NLR, and 2-h ESR increased, while lymphocyte levels decreased compared to non-recurrent patients. A comparison of the ROC curves with the sensitivity, specificity, AUC, cut-off, and asymptotic significance of CRP, fibrinogen, CA125, Fib/Alb and NLR in patients with non-recurrent and recurrent IGM are shown in Table 7 and Figure 2.

## 4. Discussion

IGM continues to be an important problem for both surgeons and patients due to the difficulty in the diagnosis and treatment of the disease, as well as acute and chronic exacerbations and long-term effects on quality of life. The prognosis of IGM is not as clear as its treatment. IGM may be misdiagnosed as breast cancer.

BC is still a worldwide public health dilemma. It is the most common female malignancy in the world and the primary cause of mortality due to cancer in women. Early diagnosis provides the best survival rate; thus, novel diagnostic markers are essential for reducing mortality and morbidity. Inflammatory factors such as the preoperative CRP, interleukin (IL)-6 and -8, and tumor necrosis factor-α are associated with BC prognosis [25,26,27]. The NLR, which is a new indicator of systemic inflammation, plays an important role in tumor progression and metastasis. However, the association between NLR and BC prognosis remains unclear.

In our study, there was a statistically significant difference between the IGM and the BC with respect to fibrinogen, Fib/Alb, CRP, WBC, neutrophil, NLR, PLR, and monocyte values. Some of these variables are components of IGM (WBC and CRP) and are, therefore, not surprising. More importantly, there was a negative relationship between CRP and albumin and a positive relationship between CRP and WBC, NLR, PLR, and ESR.

Systemic inflammation and infection can promote thrombosis by increasing the serum levels of fibrinogen, leukocytes, coagulation factors, and cytokines, and by altering the metabolism and function of certain cells, monocytes, and macrophages [28]. Although there have been many studies [29,30,31,32] on cancers and inflammatory diseases with inflammatory cytokines, fibrinogen, CRP, albumin, and hematological indexes as inflammatory markers, these parameters are limited in the differential diagnosis of IGM and early onset BC. ESR is an important indicator of acute phase response. The proteins involved in ESR include fibrinogen, alpha and beta globulin, and albumin [33]. Among these, fibrinogen has the asymmetric molecular structure that makes the largest contribution [34]. The major finding of the present study was the highest levels of fibrinogen and Fib/Alb in the IGM group. Levels of albumin, a negative acute phase reactant, were also found to be significantly lower in the IGM group than in the control group. ESR was found to be significantly increased in the IGM at both the first and second hours compared to the control. Moreover, it increased significantly in the second hour compared to BC. While the negative correlation was found between CRP and 1-h ESR with albumin, a positive correlation was found between CRP and 1-h ESR and 2-h ESR in IGM. Zheng et al. [35] found that the Fib/Alb ratio was associated with BC prognosis and claimed that the preoperative Fib/Alb ratio in particular could be related to BC survival and prognosis. Destek et al. [36] found increased plasma fibrinogen levels (460 mg/dL) in 37-year-old and 14-week-old pregnant IGM patients. β-fibrinogen-455 G > A pathological gene polymorphisms were also observed in the IGM group. As described in their case, fibrinogen gene polymorphisms and induced inflammatory and autoimmune disorders may play a role in unknown IGM etiology. Yigitbasi et al. [37] found that significant differences were not observed between the IGM, BC, and control groups with regard to total protein, albumin, and CA15-3 levels. Low-grade infections reflected by increased/decreased levels of acute phase proteins, such as fibrinogen, albumin, and Fib/Alb ratio, may be partly responsible for the inflammatory processes observed in breast lesions.

CRP, a member of the pentraxin family and a marker of inflammation, constitutes the prototype of acute phase proteins. CRP is a major acute phase reactant in humans, with an acute and rapid rise in response to infection and tissue damage. In response to all types of tissue damage and infection, the human body activates various mechanisms to correct this damage. The most important of these mechanisms is the acute phase response. In addition to radiological data and breast biopsy, circulating CRP levels may be non-invasive biomarkers that can help differentiate IGM from BC. CRP levels were found to be highest in the IGM group in the current study. A positive correlation was also found between 1-h ESR, 2-h ESR, WBC NLR, and PLR and CRP in IGM. ROC analysis results in the IGM group showed that the cut-off values for the best sensitivity and specificity for CRP were > 3.625 (94.55% and 70%, respectively). This is consistent with the results of our other study [38] and those of Jacquin-Porretaz et al. [39]. Contrary to these results, Yigitbasi et al. [37] demonstrated that the CRP levels of the BC group were significantly higher than those of the IGM and control groups. Akalın et al. [24] found that CRP values were significantly higher in breast abscess and IGM groups compared with the control group. Furthermore, there was no significant difference in CRP between the breast abscess and IGM groups. Their ROC analysis revealed cut-off values; the best sensitivity and specificity for CRP were 1.5 (61–76%) in the IGM group, suggesting that CRP cannot be a useful tool for the differential diagnosis of BC and IGM. These differences in study results may be due to the limitation of the small number of patients in all these studies. Another reason could be the sustained increase in the concentrations of acute phase proteins as the inflammation progresses. This difference may also be due to the different inflammation statuses of cancer patients.

Chronic inflammation is thought to be a predisposing factor for cancer formation by initiating the carcinogenesis process [40]. Changes in some complete blood count parameters affected by inflammation and other causes can be used to predict cancer prognosis and survival [41]. Clinicians generally use a complete blood count (CBC) in their daily practice, especially in inflammatory diseases and treatment follow-up. Recently, the use of hematological indexes, such as NLR, monocyte–to lymphocyte ratio (MLR), and PLR, as simple and inexpensive biomarkers for the demonstration of systemic inflammation, has been increasing. Çetinkaya et al. [42] investigated 41 IGM patients with a mean follow-up time of 28.4 months. These results demonstrated that an increased NLR was predictive of poor outcomes in patients with IGM. In our study, the highest levels of WBC, neutrophil, NLR, and monocytes were found in the IGM group. Moreover, a positive correlation was found between CRP and WBC, NLR, and PLR in IGM. There are positive correlations between NLR, WBC, and neutrophil in IGM. These parameters are increasingly being used as prognostic markers for predicting the prognosis of cancers [15,16,18]. Kargin et al. [43] claimed that pre-treatment NLR may be a predictive factor for long-term recurrence after treatment in patients with granulomatous mastitis. They also compared NLR in patients with (*n* = 7) and without (*n* = 52) recurrence and found that NLR was significantly higher in patients with a recurrence, but they did not perform a ROC analysis to support their observations. In our study, according to the ROC analysis for the BC and IGM groups, the sensitivity (65%) and specificity (74%) of the NLR were found to be low. According to the results of the ROC analysis, NLR might not be a good biomarker for differentiating BC and IGM. The reason for the low sensitivity and specificity may be due to the different inflammation statuses of cancer patients. The routine collection of the CBC in clinical practice is simple and cost-effective to the patient, making the NLR a highly promising indicator for monitoring the systemic inflammatory status of IGM.

Surgical treatment of IGM is controversial due to delays in wound healing, high local recurrence rates, and poor cosmetic results. In a study by Yau et al. [44], surgical treatment was applied to 11 patients diagnosed with IGM. The last surgical intervention needs of the patients occurred between 1 and 5 months. More than one surgical intervention was performed due to the recurrence of eight patients [44]. In the current study, the recurrence rate of IGM was 20% at 2 years of follow-up [44]. The relationship between the number of biochemical laboratory findings and hematological indexes and recurrence was evaluated. In patients with recurrent IGM, fibrinogen, the Fib/Alb ratio, CRP, neutrophil, NLR, and 2-h ESR increased, while lymphocyte levels decreased compared to non-recurrent patients. NLR and CRP predicted recurrence with a sensitivity of 81.2% and 100% and specificity of 61.36% and 47.73%, respectively, while the Fib/Alb ratio and fibrinogen predicted recurrence with a sensitivity of 54.55% and 72.73%, and specificity of 95.45% and 95.45%, respectively, in patients with IGM. Our results showed that these parameters have a significant effect on the recurrence of the disease. Similarly to our results, Çetinkaya et al. [42] demonstrated that increased preoperative NLR (cut-off value of 5.02) was indicated as a recurrence predictor in patients with IGM recurrence. Kargin et al. [43] reported recurrence in four (20%) patients who received steroid treatment, while recurrence was observed in three (7.9%) patients who received surgical treatment. In the study, there was no significant difference in the pre-treatment NLR and recurrence between the surgical and medical groups [43]. However, regardless of the treatment performed, the NLR of patients with relapse was higher than that of patients without relapse. Thus, the authors reported that the pre-treatment NLR value in IGM patients may predict recurrence in the long term [43]. Therefore, there is a need for large studies on the effectiveness of treatment methods applied and their effects on recurrence, based on the characteristics of patients.

The study has several limitations. Genomic whole exome or targeted exome sequencing of samples from IGM lesions and breast cancer patients could have been beneficial in identifying the difference. However, due to economic reasons, we could not perform sequencing. Most of these markers are acute phase reactants and we have aimed to check these acute phase reactants during the most acute time when these patients were newly diagnosed. Both diseases are progressive; however, we have taken samples from the patients at the time of the diagnosis at their first admission. In addition, in the results the cut off values for both diseases show a statistically significant difference. This constitutes a limitation for our study, and it would be reasonable to compare the severity of these diseases and markers according to the stage.

If inflammation is treated or self-limiting, acute phase protein levels return to normal levels within days to weeks. Therefore, acute phase proteins such as albumin, fibrinogen, and CRP can be used as markers in monitoring the treatment of IGM, which is an inflammatory disease. However, breast biopsy is the gold standard for the differential diagnosis of IGM, and is also widely used in other situations. The diagnosis of IGM is not yet supported by any serum marker. CRP, fibrinogen, and complete blood count used in daily practice, which are cheap, easy, and non-invasive, can be useful in guiding follow-up and treatment. These parameters also play a significant role in the recurrence of the disease. The relatively low number of cases in both groups is another limitation of our study. Nevertheless, our findings provide important insights for larger studies that will examine IGM patients’ responses to treatment and the effects of inflammation-induced changes in these responses.

## Figures and Tables

**Figure 1 medicina-57-00698-f001:**
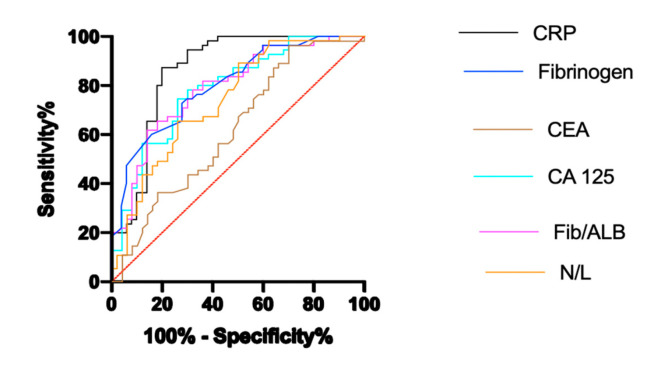
ROC analysis of parameters for breast cancer and IGM.

**Figure 2 medicina-57-00698-f002:**
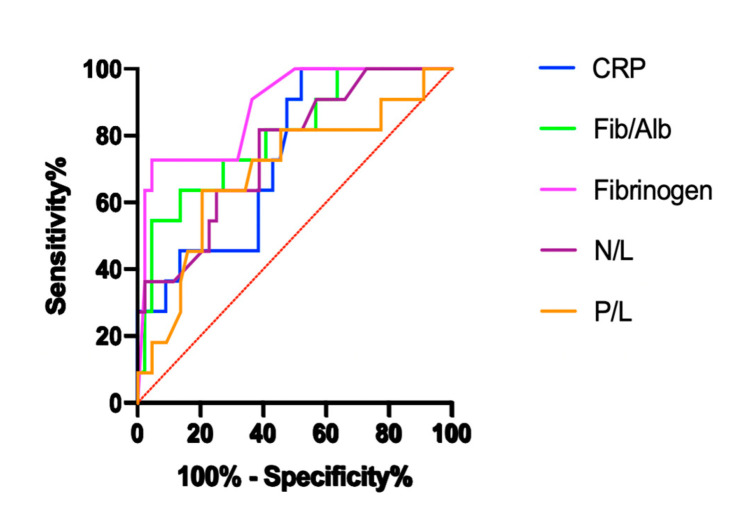
ROC analysis of parameters for patients with non-recurrent and recurrent IGM.

**Table 1 medicina-57-00698-t001:** Demographic characteristics and tumor markers of breast cancer, IGM and control groups ^a^.

	Breast Cancer (*n* = 50)	Idiopathic Granulomatous Mastitis (*n* = 55)	Control (*n* = 50)	*p*-Value ^‡^
Age (Year)	44.30 ± 7.55	43.89 ± 6.53	44.92 ± 8.02	0.773
BMI (kg/m^2^)	28.19 ± 5.66 ^b^*	26.39 ± 4.43	27.08 ± 5.30	**<0.001**
CEA (ng/mL)	2.05 ± 0.83	1.73 ± 0.81	1.77 ± 0.74	0.093
CA 19-9 (U/mL)	17.00 ± 11.81	23.29 ± 19.79	8.19 ± 3.43	**<0.001**
CA 15-3 (U/mL)	22.45 ± 13.30	19.01 ± 6.81	10.93 ± 5.23	**<0.001**
CA 125 (U/mL)	15.38 ± 8.09	27.73 ± 15.34 ^c^***	12.88 ± 7.43	**<0.001**

BMI: Body mass index; CEA: Carcinoembryonic antigen; CA: Cancer antigen; IGM: Idiopathic granulomatous mastitis. **^‡^**: Kruskal–Wallis test was used. ^a^: Control; ^b^: IGM; ^c^: Breast Cancer. *** *p* < 0.001; ** *p* < 0.01; * *p* < 0.05.

**Table 2 medicina-57-00698-t002:** Clinicopathological features of breast cancer group.

	*n*	%
Ki67 (%)	50	100 (28.16 ± 18.27)
Type of Cancer		
IDC	37	74.0
Mixed	13	26.0
*Localization*		
Right Breast	24	48.0
Left Breast	26	52.0
*Histologic Grade*		
II	34	66.0
III	16	34.0
*Estrogen Receptor*		
Negative	15	30.0
Positive	35	70.0
*Progesterone Receptor*		
Negative	16	32.0
Positive	32	68.0
*CerbB-2/ HER2/neu*		
Negative	28	56.0
Positive	22	44.0

IDC: Invasive ductal carcinoma.

**Table 3 medicina-57-00698-t003:** Biochemical laboratory findings and hematological indexes of breast cancer, IGM and control groups.

	Breast Cancer (*n* = 50)	IGM (*n* = 55)	Control (*n* = 50)	*p*-Value ^‡^
Albumin (g/dL)	4.53 ± 0.24	4.49 ± 0.29 ^a^**	4.64 ± 0.22	0.008
Total Protein (g/dL)	7.62 ± 0.31	7.25 ± 0.40	7.31 ± 0.40	0.674
Fibrinogen (mg/dL)	279.20 ± 78.39 ^a^*^.b^***	372.09 ± 77.67 ^a^**	321.12 ± 75.55	**<0.001**
Fib/Alb	61.81 ± 18.03 ^b^***	83.22 ± 18.29 ^a^***	69.30 ± 16.64	**<0.001**
CRP (mg/L)	3.24 ± 3.53 ^b^***	11.33 ± 11.00 ^a^***	1.63 ± 1.44	**<0.001**
WBC (mm^3^)	6.82 ± 2.17 ^b^***	8.62 ± 2.55 ^a^***	6.80 ± 1.69	**<0.001**
Hb (g/dL)	12.53 ± 1.36	12.07 ± 1.50 ^a^**	13.01 ± 1.12	**0.002**
Hct (%)	37.60 ± 3.88	36.49 ± 3.90 ^a^**	38.64 ± 3.09	**0.012**
PLT (mm^3^)	274.40 ± 71.82	288.13 ± 77.21	264.90 ± 58.34	0.233
NEUT (×10^3^/µL)	3.84 ± 1.40 ^b^***	5.62 ± 2.10 ^a^***	3.77 ± 1.18	**<0.001**
LYMPH (×10^3^/µL)	2.10 ± 0.77	2.07 ± 0.54	2.11 ± 0.55	0.942
NLR	1.98 ± 0.81 ^b^***	2.89 ± 1.52 ^a^***	1.90 ± 0.82	**<0.001**
PLR	150.72 ± 79.97	147.63 ± 51.84	131.31 ± 37.37	0.209
MONO (×10^3^/µL)	0.45 ± 0.18 ^b^*	0.57 ± 0.24 ^a^*	0.47 ± 0.16	**0.008**
EOS (×10^3^/µL)	0.16 ± 0.13	0.16 ± 0.12	0.15 ± 0.14	0.775
1-h ESR (mm)	18.18 ± 13.30	22.36 ± 19.59 ^a^**	11.86 ± 4.93	**0.001**
2-h ESR (mm)	42.40 ± 22.73 ^a^*	47.31 ± 28.58 ^a^***	29.44 ± 9.70	**<0.001**

**Fib/Alb:** Fibrinogen/Albumin; **CRP:** C-reactive protein; **WBC:** White blood cells; **Hb:** Hemoglobin; **Hct:** Hematocrit; **PLT**: Platelets; **NEUT:** Neutrophil; **LYMPH:** Lymphocyte; **NLR:** Neutrophil–lymphocyte ratio; **PLR:** Platelet–lymphocyte ratio; **MONO:** Monocyte; **EOS:** Eosinophil; **ESR:** erythrocyte sedimentation rate; **IGM:** Idiopathic granulomatous mastitis. ^‡^: Kruskal–Wallis test was used. ^a^: Control; ^b^: IGM; ^c^: Breast cancer. *** *p* < 0.001; ** *p* < 0.01; * *p* < 0.05.

**Table 4 medicina-57-00698-t004:** Correlation analysis in IGM patients.

	r	*p*
**CRP-Albumin**	−0.411	0.002
**CRP-1-h ESR**	0.571	<0.001
**CRP-2-h ESR**	0.481	<0.001
**CRP-WBC**	0.269	0.047
**CRP-NLR**	0.314	0.019
**CRP-PLR**	0.288	0.033
**Fib-Fib/Alb**	0.960	<0.001
**Alb-1-h ESR**	−0.383	0.004
**1-h ESR- 2-h ESR**	0.908	<0.001
**1-h ESR-PLR**	0.408	0.002
**WBC-NLR**	0.302	0.033
**WBC-NEUT**	0.682	<0.001
**NEUT-NLR**	0.548	<0.001

**IGM:** Idiopathic granulomatous mastitis; **Fib/Alb:** Fibrinogen/Albumin; **CRP:** C-Reactive Protein; **WBC:** white blood cells; **NLR:** neutrophil–lymphocyte ratio; **PLR:** platelet–lymphocyte ratio; **ESR:** erythrocyte sedimentation rate; **NEUT:** neutrophil.

**Table 5 medicina-57-00698-t005:** Comparison of CRP, fibrinogen, CA125, Fib/Alb and NLR levels using ROC analysis for breast cancer and IGM groups.

	Area under Curve (%CI)	Sensitivity (%)	Specificity (%)	Cut-Off (<Value)	*p*-Value
**CRP (mg/L)**	0.862	94.55	70	>3.625	<0.001
**Fibrinogen (mg/dL)**	0.801	72.73	72	>345.0	<0.001
**CA 125 (U/mL)**	0.788	80	66	>15.47	<0.001
**Fib/Alb**	0.792	61.82	86	>82.79	<0.001
**NLR**	0.742	65.45	74	>2.285	<0.001

**CA:** Cancer antigen; **CRP:** C-reactive protein; **Fib/Alb:** Fibrinogen/Albumin. **NLR:** Neutrophil–lymphocyte ratio.

**Table 6 medicina-57-00698-t006:** Biochemical laboratory findings and hematological indexes of patients with non-recurrent and recurrent IGM.

	Non-Recurrent IGM (*n* = 44)	Recurrent IGM (*n* = 11)	*p*
**Albumin (g/dL)**	4.48 ± 0.30	4.22 ± 1.00	0.064
**Total Protein (g/dL)**	7.23 ± 0.40	6.86 ± 1.65	0.088
**Fibrinogen (mg/dL)**	351.36 ± 69.07	395.03 ± 110.77	**0.037**
**Fib/Alb**	79.25 ± 16.96	88.87 ± 25.42	**0.049**
**CRP (mg/L)**	9.24 ± 7.50	15.98 ± 14.43	**0.011**
**WBC (mm^3^)**	8.58 ± 2.52	8.36 ± 2.66	0.388
**Hb (g/dL)**	11.93 ± 1.54	11.85 ± 2.74	0.444
**Hct (%)**	36.19 ± 4.13	35.51 ± 8.09	0.336
**PLT (mm^3^)**	284.36 ± 79.59	273.59 ± 77.28	0.320
**NEUT (×10^3^/µL)**	5.23 ± 1.56	6.33 ± 2.95	**0.034**
**LYMPH (×10^3^/µL)**	2.13 ± 0.53	1.84 ± 0.57	**0.034**
**NLR**	2.51 ± 0.77	3.59 ± 2.32	**0.005**
**PLR**	139.85 ± 47.35	153.93 ± 60.39	0.174
**MONO (×10^3^/µL)**	0.53 ± 0.17	0.61 ± 0.35	0.109
**EOS (×10^3^/µL)**	0.17 ± 0.12	0.13 ± 0.12	0.123
**1-h ESR (mm)**	23.98 ± 20.54	50.45 ± 29.48	0.105
**2-h ESR (mm)**	17.27 ± 10.02	37.00 ± 16.04	**0.042**

**Fib/Alb:** Fibrinogen/Albumin; **CRP:** C-reactive protein; **WBC**: White Blood Cells; **Hb**: Hemoglobin; **Hct:** Hematocrit; **PLT**: Platelets; **NEUT:** Neutrophil; **LYMPH:** Lymphocyte; **NLR:** Neutrophil–lymphocyte ratio; **PLR:** Platelet–lymphocyte ratio; **MONO:** Monocyte; **EOS:** Eosinophil.

**Table 7 medicina-57-00698-t007:** Comparison of CRP, Fib/Alb, fibrinogen, and NLR levels using ROC analysis for patients with non-recurrent and recurrent IGM.

	CRP	Fib/Alb	Fibrinogen	NLR	PLR
**AUC**	0.7366	0.7996	0.8843	0.7572	0.6963
**Std. Error**	0.07641	0.07549	0.05496	0.07854	0.09402
**95% CI**	0.5868 to 0.8863	0.6516 to 0.9475	0.7766 to 0.9920	0.6033 to 0.9112	0.5120 to 0.8806
***p* value**	0.016	0.0023	<0.0001	0.0088	0.0456
**Cut-off value**	>7.280	>105.0	>430.0	>2.570	>170.0
**Sensitivity (%)**	**100**	54.55	72.73	**81.82**	63.64
**Specificity (%)**	47.73	**95.45**	**95.45**	61.36	79.55

**CRP:** C-reactive protein; **Fib/Alb:** Fibrinogen/Albumin; **NLR:** Neutrophil–lymphocyte ratio; **PLR:** Platelet–lymphocyte ratio; **AUC:** area under curve.

## Data Availability

The data sets analyzed during the current study are available from the corresponding author on reasonable request.

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
