# Peer review of "Preoperative Fibrinogen and Hematological Indexes in the Differential Diagnosis of Idiopathic Granulomatous Mastitis and Breast Cancer"

_medicina, 2021, doi:10.3390/medicina57070698_

Round 1
Reviewer 1 Report
In order to discover a clinical diagnostic biomarker for distinguishing between Idiopathic granulomatous mastitis (IGM) and breast cancer, they should perform whole exome or targeted exome sequencing of samples from IGM legions and breast cancer patients. By doing so, the authors can shed a fresh light on what kind of genomic alterations underlie the difference between IGM and breast cancer pathologies. If the authors could not perform such sequencing effort in their investigation, make explanations of their results' limitations (based on the above comment) in the revised manuscript's discussion section.
Author Response
The limitation of the study was added to the discussion section according to direction requested by the Reviewer.

Reviewer 2 Report
The manuscript “Preoperative Fibrinogen and Hematological Indexes in the Differential Diagnosis of Idiopathic Granulomatous Mastitis and Breast Cancer” by Velidedeoglu et al. discusses the parameters that can be used for diagnosis of IGM from breast cancer. The concept of the manuscript is interesting and overall, manuscript is well written and authors draw logical conclusions. Following critiques are mentioned-
- One main point that authors have not addressed is that IGM and breast cancer, are diseases that are progressive in nature involving differential microenvironment and secretory properties at various stages of progression. How can one set of markers predict the diagnosis of disease which may be at different stages in real life in different patients?
- Authors can expand the conclusions to discuss the potential flaws of their study and how they can diagnose different patients uniformly even though they may represent different stages of disease.
Author Response
Most of these markers are acute phase reactants and we have aimed to check these acute phase reactants during the most acute time when these patients were newly diagnosed. We have taken these samples when these patients were just diagnosed. Both diseases are progressive; however, we have taken samples from the patients at the time of the diagnosis at their first admission. Also, in the results the cut off values for both diseases show a statistically significant difference. This would be a limitation for our study and it would be reasonable to compare the severity of these diseases and markers according to the stage.
